# Impact of rice *GENERAL REGULATORY FACTOR14h* (*GF14h*) on low-temperature seed germination and its application to breeding

Yusaku Sugimura[1], Kaori Oikawa[1], Yu Sugihara[2¤a], Hiroe Utsushi[1], Eiko Kanzaki[1], Kazue Ito[1], Yumiko Ogasawara[1], Tomoaki Fujioka[3], Hiroki Takagi[1¤b], Motoki Shimizu[1], Hiroyuki Shimono[4,5], Ryohei Terauchi[1,2], Akira Abe[1] *

**1** Iwate Biotechnology Research Center, Kitakami, Iwate, Japan, **2** Crop Evolution Laboratory, Kyoto University, Muko, Kyoto, Japan, **3** Iwate Agricultural Research Center, Kitakami, Iwate, Japan, **4** Faculty of Agriculture, Iwate University, Morioka, Iwate, Japan, **5** Agri-Innovation Center, Iwate University, Morioka, Iwate, Japan

¤a Current address: The Sainsbury Laboratory, University of East Anglia, Norwich, United Kingdom
¤b Current address: Faculty of Bioresources and Environmental Science, Ishikawa Prefectural University, Nonoichi, Ishikawa, Japan

* a-abe@ibrc.or.jp

**Data Availability Statement:** The NGS raw reads were deposited in the DNA Data Bank of Japan (DDBJ) under the BioProject accession numbers

## Abstract

Direct seeding is employed to circumvent the labor-intensive process of rice (*Oryza sativa*) transplantation, but this approach requires varieties with vigorous low-temperature germination (LTG) when sown in cold climates. To investigate the genetic basis of LTG, we identified the quantitative trait locus (QTL) *qLTG11* from rice variety Arroz da Terra, which shows rapid seed germination at lower temperatures, using QTL-seq. We delineated the candidate region to a 52-kb interval containing *GENERAL REGULATORY FACTOR14h* (*GF14h*) gene, which is expressed during seed germination. The Arroz da Terra *GF14h* allele encodes functional GF14h, whereas Japanese rice variety Hitomebore harbors a 4-bp deletion in the coding region. Knocking out functional *GF14h* in a near-isogenic line (NIL) carrying the Arroz da Terra allele decreased LTG, whereas overexpressing functional *GF14h* in Hitomebore increased LTG, indicating that *GF14h* is the causal gene behind *qLTG11*. Analysis of numerous Japanese rice accessions revealed that the functional *GF14h* allele was lost from popular varieties during modern breeding. We generated a NIL in the Hitomebore background carrying a 172-kb genomic fragment from Arroz da Terra including *GF14h*. The NIL showed superior LTG compared to Hitomebore, with otherwise comparable agronomic traits. The functional *GF14h* allele from Arroz da Terra represents a valuable resource for direct seeding in cold regions.

## Author summary

Rice serves as a fundamental crop sustaining over half of the global population. With the rapid growth of the world's population, it will become increasingly important to improve rice productivity. On the other hand, the aging of rice farmers in Japan has resulted in a

PRJDB13449, PRJDB13450, PRJDB13864, and PRJDB17450. Supplementary Tables S2 and S5 include the SRA accession numbers. The genome sequences assembled in this study were deposited at Zenodo (https://doi.org/10.5281/zenodo.10460309). All relevant data are within the paper and its Supporting Information files.

**Funding:** This work was partly supported by a grant from the Ministry of Agriculture, Forestry, and Fisheries of Japan (Genomics-based Technology for Agricultural Improvement, IVG3007), received by HT, RT and AA. The funders had no role in study design, data collection and analysis, decision to publish, or preparation of the manuscript.

**Competing interests:** The authors have declared that no competing interests exist.

constant labor shortage. To address this, direct seeding, in which seeds are sown directly in rice fields without going through the most labor-intensive part of the rice cultivation process, i.e., seedling production and transplanting, has been recommended. However, prevalent elite rice varieties are known to be unsuitable for direct seeding due to their poor seed germination ability under low-temperature conditions. In this study, we show for the first time that *GF14h* gene from the Portuguese variety Arroz da Terra improves seed germination at low temperatures (LTG). In addition, a novel cross-bred line was generated by introducing the *GF14h*-containing genomic segment from Arroz da Terra into Hitomebore, a widely cultivated variety in northern Japan. This line is expected to be used as a pre-breeding material to enhance LTG. This study will provide a genetic basis for LTG and contribute to basic and applied research progress.

## Introduction

Low-temperature seed germination (LTG) is a pivotal agronomic trait in rice (*Oryza sativa*). As rice originated from tropical and subtropical regions, it is highly susceptible to low-temperature conditions compared to other cereal crops such as wheat (*Triticum aestivum*) and barley (*Hordeum vulgare*) [1]. Nevertheless, rice is produced in temperate and high-altitude regions, where it frequently experiences temperatures below 20°C. In Japan, rice is abundantly cultivated in relatively cold areas such as Tohoku and Hokkaido. In recent years, there has been an increasing demand to shift from conventional transplantation-based rice cultivation to direct seeding to reduce labor and costs. However, direct seeding raises the risk of exposure to low temperatures during seed germination [2]. Therefore, to expand the use of direct seeding, it is crucial to breed rice cultivars with enhanced LTG.

LTG is a quantitative trait regulated by complex molecular mechanisms. Linkage mapping and genome-wide association studies (GWAS) have identified over 30 LTG-related quantitative trait loci (QTLs) or genomic regions associated with this trait, located on all 12 rice chromosomes [3–20]. However, only a few genes involved in LTG have been described, such as *qLTG3-1* [3] and *STRESS-ASSOCIATED PROTEIN16* (*OsSAP16*) [15]. The *qLTG3-1* gene, encoding a protein of unknown function, has a substantial influence on LTG [3]. During seed germination, *qLTG3-1* expression is strongly induced in embryos, which leads to the loosening of the tissues covering the embryo by promoting vacuolation [3]. *OsSAP16* encodes a stress-associated protein with two AN1-C2H2 zinc finger domains [15]. OsSAP16 presumably acts as a regulator of LTG.

14-3-3 proteins are regulatory proteins that are widely conserved in eukaryotes. These proteins bind to phosphorylated serine and tyrosine residues in their target proteins that participate in signal transduction and the regulation of gene expression [21, 22], thus altering their enzymatic activity, subcellular localization, stability, or protein–protein interactions [23–25]. The rice genome encodes eight 14-3-3 proteins, named GF14a–h for GENERAL REGULATORY FACTOR14 [26]. *GF14h* is involved in rice seed germination under optimal temperature conditions [27,28]. In addition, GF14h contributes to phytohormone signaling, including abscisic acid and gibberellin signaling [27,28]. However, it remains unclear whether *GF14h* promotes seed germination under low-temperature conditions [28].

QTL pyramiding has been proposed as a breeding concept [29] for bringing together several QTLs (or genes) related to agronomically important traits in the genetic background of locally adapted elite cultivars. In practice, it is essential to generate pre-breeding materials for QTL pyramiding, i.e., near-isogenic lines (NILs) that harbor one or a few genomic segments

introgressed from the donor parent into the genome of the recipient parent through a combination of continuous backcrossing and selfing via marker-assisted selection [30]. In this study, we determined that *GF14h* is responsible for an LTG-related QTL in Portuguese rice variety Arroz da Terra. We generated a NIL in the background of rice cultivar Hitomebore, which is adapted for growth in northern Japan, by replacing its *GF14h* genomic fragment with that from Arroz da Terra and tested its LTG performance.

## Results

### Evaluation of QTLs associated with low-temperature germination using the Portuguese rice variety Arroz da Terra

We investigated seed germination characteristics of a Portuguese rice variety Arroz da Terra and a Japanese varieties Iwatekko and Hitomebore. Under low-temperature conditions (15°C), Arroz da Terra exhibited significantly higher germination rates compared to Iwatekko and Hitomebore during days 7−15, particularly showing a 30−40% increase in germination rate 10−11 days after imbibition (Fig 1A and 1B). Similarly, at normal temperature conditions (25°C), Arroz da Terra showed superior germination rates 2−4 days after imbibition, especially with a 30% difference observed 3 days after imbibition (Fig 1A and 1C). These findings indicate that Arroz da Terra exhibits more vigorous germination under normal and low-temperature conditions than the Japanese cultivars Iwatekko and Hitomebore. To identify the genes responsible for this difference, we searched for QTLs involved in the high LTG of Arroz da Terra. We previously generated a set of 200 RILs at the $F_7$ generation derived from a cross between Arroz da Terra and Iwatekko (S1 Fig) [31]. We phenotyped all RILs for LTG at 13°C and selected the 20 RILs with the highest LTG and the 20 RILs with the lowest LTG. We assembled two pools of seedlings with low or high LTG and extracted their genomic DNA for whole-genome sequencing on the Illumina platform (S1 Fig) [31]. We mapped the resulting sequencing reads to the Nipponbare rice reference genome (IRGSP-1.0) and performed QTL-seq analysis using our new high-performance pipeline [32]. Based on the Δ(SNP-index), we identified three QTLs related to LTG on chromosome 3 (*qLTG3-1* and *qLTG3-2*) and chromosome 11 (*qLTG11*) (Fig 1D), which is consistent with the results of a previous study [31].

The *qLTG3-1* region contained the gene Os03g0103300, which was reported to be involved in LTG in a study using rice cultivar Italica Livorno, which has high LTG, and Hayamasari, which has low LTG [3]. An examination of its coding sequences in Arroz da Terra, as well as Iwatekko and Hitomebore, revealed that they were identical to those found in Italica Livorno and Hayamasari, respectively (S2 Fig). While Italica Livorno harbored a functional haplotype for this gene, Hayamasari carried a loss-of-function haplotype due to a 71-bp deletion (S2 Fig) [3]. Therefore, we propose that the causal gene for the QTL *LTG3-1* is Os03g0103300.

To evaluate the contribution of the two other QTLs to LTG, we generated NILs harboring a segment from the Arroz da Terra genome for each QTL (approximately 5 Mb) in the Hitomebore background (Figs 1E, S3A and S4A). We detected no clear effect of *qLTG3-2* on LTG, as *qLTG3-2*-NIL and Hitomebore showed similar seed germination rates at 15°C (S4B Fig). By contrast, *qLTG11*-NIL showed a significantly higher rate of germination than Hitomebore 7−12 days after imbibition at 15°C, particularly after 8 and 9 days, showing a difference of more than 40% (Fig 1E and 1F), indicating that *qLTG11* enhances LTG. *qLTG11*-NIL seeds also germinated more rapidly than Hitomebore seeds under normal conditions (25°C), although with a smaller difference between the two genotypes than at low temperature (S5 Fig). We therefore focused our analysis on *qLTG11*.

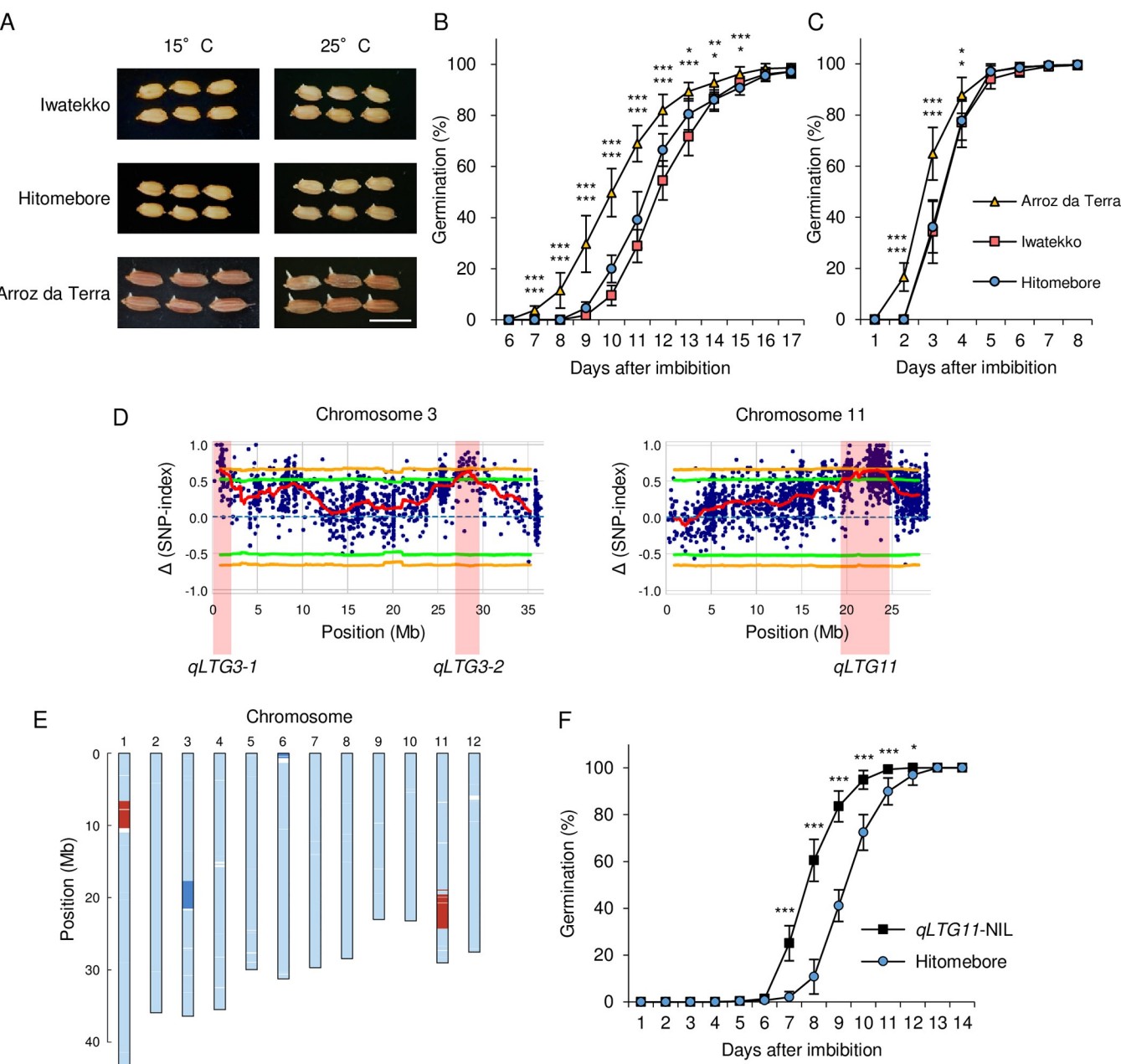

**Fig 1. Effects of quantitative trait loci (QTLs) on low-temperature seed germinability.** (A) Representative photographs showing the germination of seeds from the Iwatekko, Hitomebore, and Arroz da Terra varieties 11 days (15°C) or 3 days (25°C) after the onset of seed imbibition. Scale bar, 1 cm. (B–C) Germination time courses of Iwatekko, Hitomebore, and Arroz da Terra at 15°C (B) or 25°C (C). Values are means ± standard deviation (SD) from biologically independent samples ($n = 8$). Dunnett's test shows significant differences in germination for Hitomebore (upper) and Iwatekko (lower) compared with Arroz da Terra at each time point (*$P < 0.05$, **$P < 0.01$ and ***$P < 0.001$). (D) Map positions of QTLs for low-temperature germination, as determined by QTL-seq. The Δ (SNP-index) values (red lines) were plotted for chromosomes 3 and 11, with statistical confidence intervals under the null hypothesis of no QTL (green, $P < 0.05$; orange, $P < 0.01$). (E) Diagram showing the genotype of *qLTG11*-NIL. *qLTG11*-NIL harbors the Arroz da Terra allele at *qLTG11* on chromosome 11. Light blue indicates genomic fragments from Hitomebore; red indicates genomic fragments from Arroz da Terra; dark blue indicates heterozygous regions. (F) Germination time courses of Hitomebore and *qLTG11*-NIL at 15°C. Values are means ± SD from biologically independent samples ($n = 10$). Two-tailed t-test was used between *qLTG11*-NIL and Hitomebore for each time point (*$P < 0.05$ and ***$P < 0.001$).

## Identification of *GF14h* as the candidate gene for *qLTG11*

To delineate the *qLTG11* region, we carried out map-based cloning using a segregating population derived from a cross between BC$_2$F$_3$ line *qLTG11*-NIL and Japanese elite cultivar Hitomebore (S3A Fig). For mapping, we conducted germination tests at 15˚C. We narrowed down the genomic region containing the QTL to a 52-kb segment (from 23.512 Mb to 23.564 Mb) on chromosome 11 based on the Nipponbare reference genome (IRGSP-1.0) (Fig 2A). This interval contains two annotated genes based on the Nipponbare genome sequence (Fig 2B). We compared the genomic sequence of Hitomebore and Arroz da Terra across the candidate region using *de novo* genome assembly obtained from Nanopore long reads. The cultivars Hitomebore and Nipponbare had an identical genomic sequence over the entire candidate region (S6A Fig). By contrast, the genome sequence from Arroz da Terra was substantially different from that of Nipponbare, with the equivalent candidate region spanning approximately 94 kb (Figs 2B and S6B).

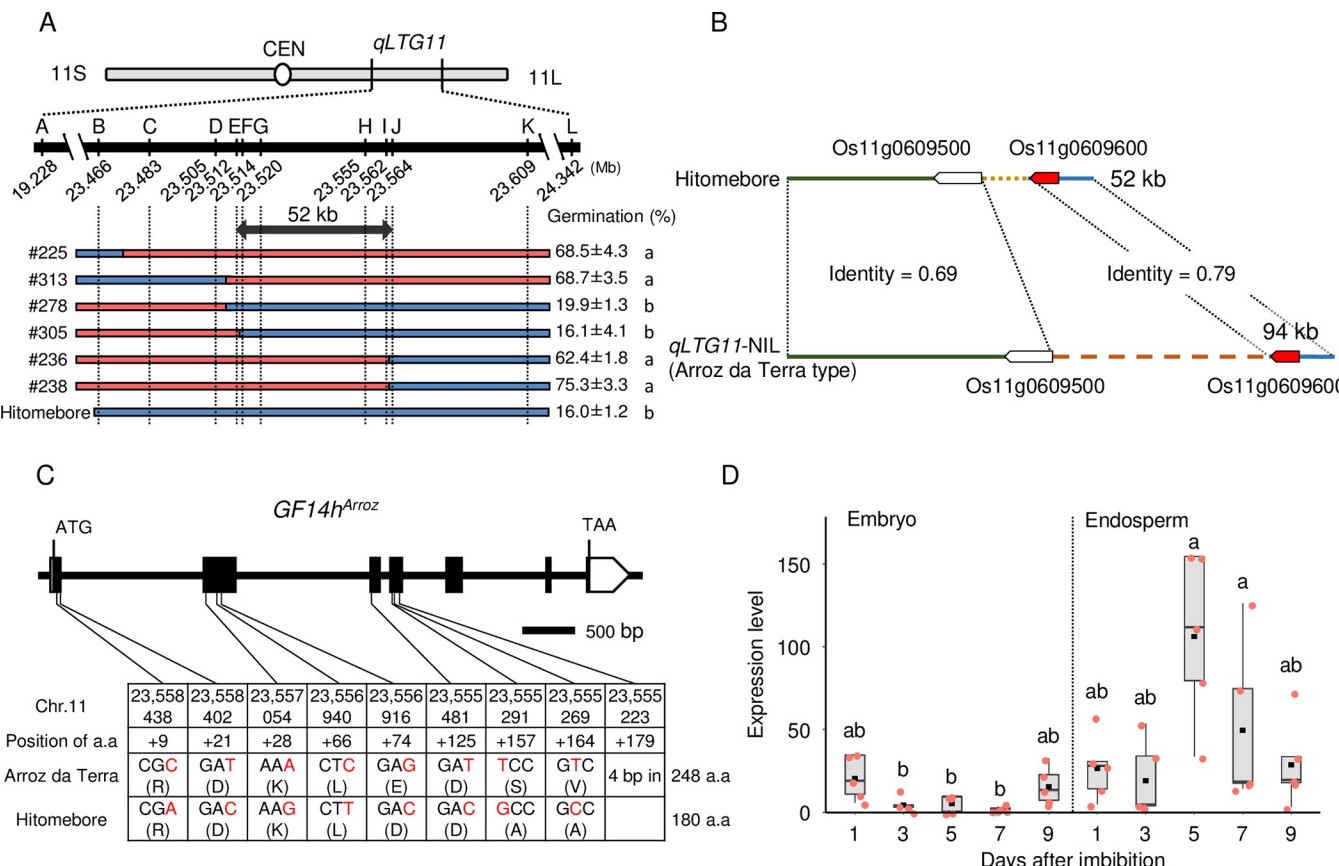

**Fig 2. Positional cloning of *qLTG11*.** **(A)** Fine mapping of *qLTG11* to a 52-kb region between markers E and J. The chromosomal positions are based on the Nipponbare reference genome (Os-Nipponbare-Reference-IRGSP-1.0). Germination percentage was determined at 9 days of incubation at 15˚C. Red and blue rectangles indicate chromosomal segments homozygous for Arroz da Terra or Hitomebore, respectively. Different lowercase letters indicate significant differences (*n* = 3 biologically independent samples, *P* < 0.001, Tukey's HSD test). **(B)** Genomic structure of the candidate genomic region in Arroz da Terra and Hitomebore. Os11g0609600 (shown in red), encoding GF14h, is expressed in germinating seeds. **(C)** Diagram of the *GF14h* gene structure and sequence polymorphisms between Arroz da Terra and Hitomebore. The chromosomal positions are based on the Nipponbare reference genome. The coding region of *GF14h* in Hitomebore is identical to that in Nipponbare. The 4-bp deletion in Hitomebore causes a frameshift and the introduction of a premature stop codon. **(D)** Relative *GF14h* expression levels in germinating seeds of *qLTG11*-NIL. This expression analysis was conducted by RT-qPCR. In the boxplots, the box edges represent the upper and lower quantiles, the horizontal line in the middle of the box represents the median value, whiskers represent the lowest quantile to the top quantile, and the black squares show the mean. Five biological replicates were measured independently. Different lowercase letters indicate significant differences based on Tukey's HSD test (*P* < 0.05). *OsActin1* (Os03g0718100) was used for normalization.

As the causal gene behind the variation in LTG is likely expressed in seeds, we performed transcriptome deep sequencing (RNA-seq) during seed germination in Hitomebore and *qLTG11*-NIL (S1 Table). Within the candidate region, the gene Os11g0609600, corresponding to the *14-3-3* gene *GF14h*, was expressed in both Hitomebore and *qLTG11*-NIL, whereas Os11g0609500 (*Jacalin-like lectin domain containing protein*) was not expressed in seeds (S7 Fig), thus suggesting that *GF14h* is a strong candidate gene for LTG. The *GF14h* gene structure and haplotypes in Arroz da Terra and Hitomebore are shown in Fig 2C. We detected a 4-bp deletion in the *GF14h* coding region in Hitomebore, causing a frameshift mutation predicted to introduce a premature stop codon (Figs 2C and S8). These results suggest that Hitomebore carries a loss-of-function allele of *GF14h*. To assess the role of *GF14h* in LTG, we examined the expression pattern of the putative functional *GF14h* (*GF14h$^{Arroz}$*) allele during seed germination at low temperature (15˚C) using *qLTG11*-NIL. RT-qPCR analysis of *GF14h* expression levels showed that they were comparable in the embryo and endosperm at 1 and 3 days after the onset of seed imbibition (Fig 2D). At the beginning of germination, when a white coleoptile was visible (5 and 7 days after seed imbibition), *GF14h* expression levels rose in the endosperm, but not in the embryo (Fig 2D). By nine days of imbibition, when most seeds had germinated, *GF14h* expression in the endosperm returned to basal levels (Fig 2D). These results support the notion that *GF14h* plays a role in seed germination at low temperature.

## *GF14h* plays a vital role in LTG

To investigate the contribution of GF14h to LTG, we knocked out the functional *GF14h* copy present in *qLTG11*-NIL by clustered regularly interspersed short palindromic repeat (CRISPR)/CRISPR-associated nuclease 9 (Cas9)-mediated gene editing and evaluated LTG. Specifically, we introduced two single guide RNA (sgRNA) constructs targeting the exons of *GF14h* individually into *qLTG11*-NIL by Agrobacterium-mediated transformation. We chose to knock out *GF14h* in the *qLTG11*-NIL background rather than Arroz da Terra to evaluate the specific contribution of *GF14h* to LTG without the influence of *qLTG3-1*, which would be present in the Arroz da Terra background. To accurately evaluate the phenotypes of the edited plants, we selected heterozygous plants in the T$_0$ generation and isolated homozygous mutant lines and their unedited homozygous siblings in the T$_1$ generation. We obtained four knockout lines (*gf14h-1*, *gf14h-2*, *gf14h-3*, and *gf14h-4*) and their wild-type sibling (WT$^{Arroz}$) (S9 Fig). We detected significant drops in the germination percentage in all four knockout lines compared to WT$^{Arroz}$ (Fig 3A and 3B). We also found that the knockout lines tended to have lower germination rates than WT$^{Arroz}$ under normal temperature conditions (25˚C) (S10 Fig), which is consistent with the previous report [27]. Furthermore, we generated transgenic lines in the Hitomebore background overexpressing the functional *GF14h* allele from Arroz da Terra under the control of the CaMV 35S promoter. In these overexpression lines, *GF14h* expression increased approximately 1,000-fold compared to the wild-type sibling (WT$^{Hitomebore}$) (S11 Fig). Importantly, the overexpression lines showed higher LTG than WT$^{Hitomebore}$ when tested at 15˚C (Fig 3C and 3D). Taken together, these data indicate that *GF14h* is a key gene involved in LTG.

## Loss-of-function alleles *GF14h* and *qLTG3-1* increased in frequency during rice breeding in Japan

We reconstructed the *GF14h* haplotype network using genotype data obtained from whole-genome resequencing of 492 *O. sativa* accessions from various collections, including the World Rice Core Collection [33], the Rice Core Collection of Japanese Landraces [34], and a set of Japanese landraces and modern varieties [35], in addition to 11 wild rice (*O. rufipogon*) accessions [36] (S2 Table). We distinguished 10 haplotypes for the *GF14h* coding region based

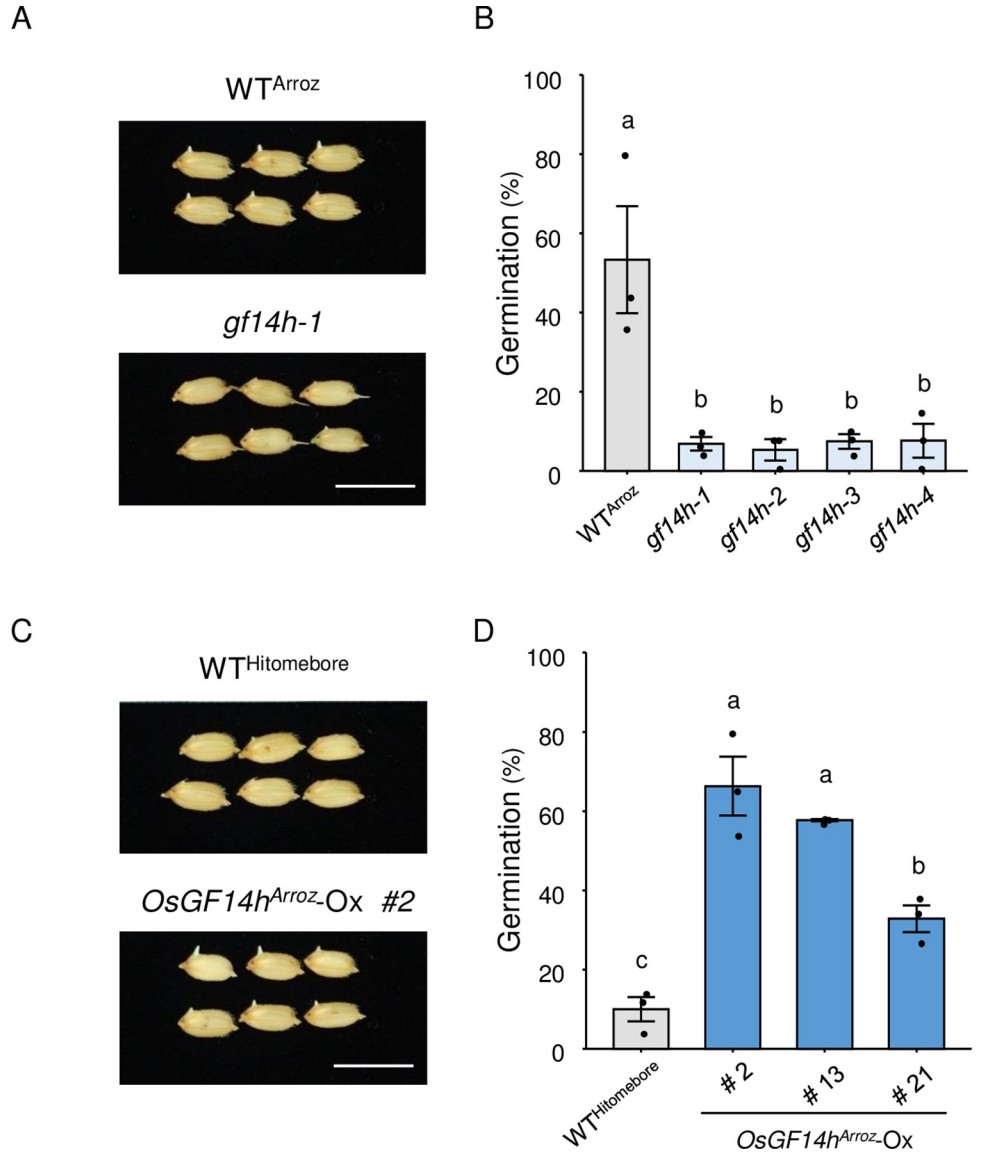

**Fig 3. Effect of *GF14h* mutation and overexpression on low-temperature germination. (A)** Representative photographs showing seed germination in wild-type harboring Arroz-type *GF14h* (WT^Arroz) and CRISPR/Cas9 knockout lines (*gf14h-1*) at 8 days after the onset of seed imbibition. Scale bar, 1 cm. **(B)** Seed germination rate of WT^Arroz and its CRISPR/Cas9 knockout lines at 7 days of seed imbibition at 15˚C. The two target constructs (S9 Fig) were introduced into the *qLTG11*-NIL line. Data are means ± standard error (SE, *n* = 3). Different lowercase letters indicate significant differences based on Tukey's HSD test (*P* < 0.01). **(C)** Representative photographs showing seed germination of wild-type (WT^Hitomebore) and *OsGF14h^Arroz* overexpression lines (*OsGF14h^Arroz*-Ox #2) at 7 days of seed imbibition at 15˚C. Scale bar, 1 cm. **(D)** Seed germination rate of WT^Hitomebore and *GF14h^Arroz* overexpression lines in the Hitomebore background at 7 days of seed imbibition at 15˚C. Data are means ± SE (*n* = 3). Different lowercase letters indicate significant differences based on Tukey's HSD test (*P* < 0.05).

on 11 polymorphic sites comprising one frameshift mutation caused by a 4-bp deletion, four nonsynonymous single nucleotide polymorphisms (SNPs), and six synonymous SNPs (S3 Table). The conversion of the functional allele Hap2 to its nonfunctional allele Hap1 required only a single step: a 4-bp deletion (S12 Fig).

We analyzed the haplotype frequencies of *GF14h* and *qLTG3-1* in Japanese landraces and cultivars, which we grouped according to their time of release. More than half of all Japanese

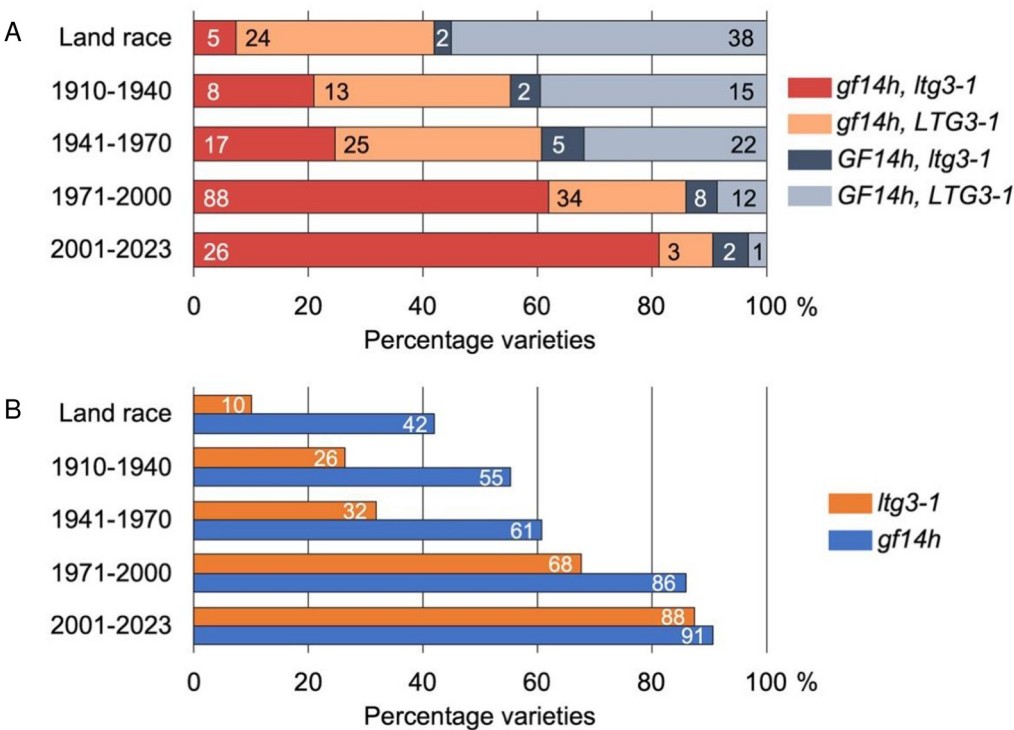

**Fig 4. Gradual selection of loss-of-function alleles in *GF14h* and *qLTG3-1* during rice breeding in Japan.** A total of 350 Japanese varieties were examined, including the World Rice Core Collection [33], the Rice Core Collection of Japanese Landraces [34], and the collection of Japanese core cultivars [35]. The allele type at each gene was determined to be functional or nonfunctional by the k-mer method using Illumina short reads for each variety. **(A)** Proportion of allele type combinations at *GF14h* and *qLTG3-1* sorted by breeding year. **(B)** Proportion of nonfunctional allele types at *GF14h* and *qLTG3-1* sorted by breeding year.

landraces carried functional alleles of both *GF14h* and *LTG3-1* (Fig 4A). The next most common allele combination among the Japanese landraces was a nonfunctional *GF14h* allele with a functional *LTG3-1* allele (Fig 4A). The percentage of lines with loss-of-function alleles at both *GF14h* and *LTG3-1* has increased since the beginning of crossbreeding in Japan in the early 20th century, with more than 80% of varieties released after 2001 carrying loss-of-function alleles for both genes (Fig 4A and 4B). Looking at each gene separately in landraces, only a few lines carried a loss-of-function allele for *LTG3-1*, whereas roughly half of all lines already harbored a loss-of-function allele for *GF14h* (Fig 4B). Modern breeding thus appears to have increased the proportion of loss-of-function alleles for these two genes, with a substantial increase in *LTG3-1*, reaching almost 90% among lines bred after 2001 (Fig 4B).

## The Arroz da Terra *GF14h* allele could be valuable for rice breeding

To assess how useful the above findings might be to practical breeding programs, we developed new breeding materials. QTL pyramiding, a strategy for introducing multiple QTLs for desired traits into a single genetic background, is a key strategy employed in current breeding. An essential step in QTL pyramiding is the generation of NILs containing the desired QTLs. Therefore, we developed a NIL, termed NIL-*GF14h^{Arroz}*, using the elite cultivar Hitomebore as the genetic background into which we introgressed a 172-kb region from the Arroz da Terra genome containing *GF14h* (S3 Fig). This NIL showed a higher seed germination rate under low-temperature conditions compared to Hitomebore (Fig 5A and 5B). In addition, although no significant difference was observed, it is likely that NIL-*GF14h^{Arroz}* tends to be more

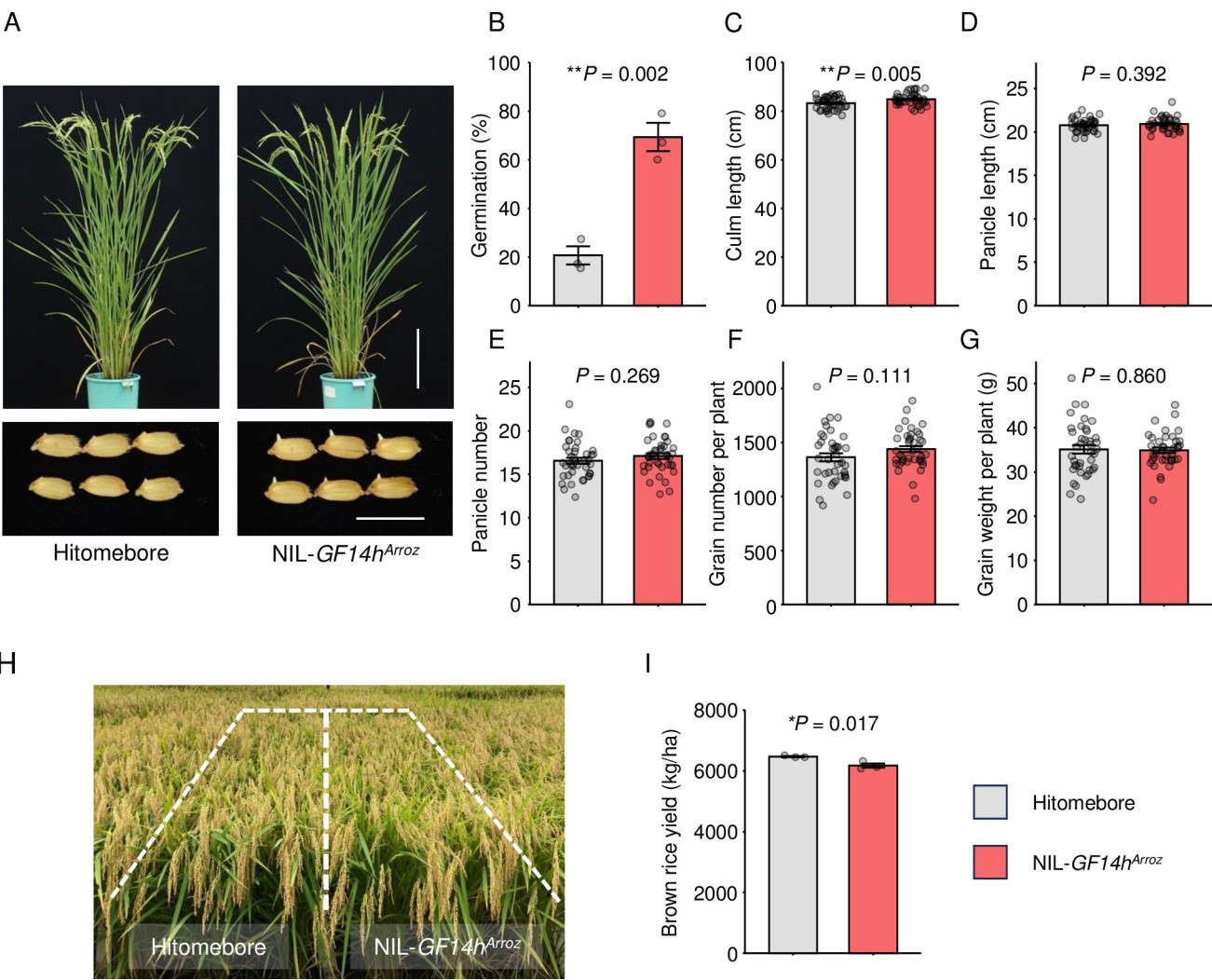

**Fig 5. Phenotypic analysis of a near-isogenic line homozygous for Arroz-type *GF14h* in the Hitomebore genetic background (NIL-*GF14h^Arroz^*). (A)** Gross morphology of Hitomebore and NIL-*GF14h^Arroz^* at 87 days after transplanting (top row), and seed germination at 9 days after the onset of seed imbibition at 15°C (bottom row). Scale bars, 20 cm (top), 1 cm (bottom). **(B)** Low-temperature germination ability of NIL-*GF14h^Arroz^* at 8 days after the onset of seed imbibition at 15°C. Values are means ± SE of biologically independent samples (*n* = 3). Asterisks indicate significant differences, as determined by two-tailed *t*-test. **(C–G)** Agronomic traits in Hitomebore and NIL-*GF14h^Arroz^*: culm length (**C**), panicle length (**D**), panicle number (**E**), grain number per plant (**F**), and grain weight per plant (**G**). Values are means ± SE of biologically independent plants (*n* = 40). **(H)** Side view of Hitomebore and NIL-*GF14h^Arroz^* growing in the field under typical rice-growing conditions at the maturity stage. **(I)** Brown rice yield per unit area. Values are means ± SE of biologically independent plots (*n* = 3). *P*-values calculated by *t*-tests are listed throughout the figure.

susceptible to pre-harvest sprouting than Hitomebore (S13 Fig). Importantly, we observed no substantial differences in five agronomic traits (culm length, panicle length, panicle number, grain number, and grain weight) between NIL-*GF14h^Arroz^* and Hitomebore (Fig 5C–5G). Brown rice yield was slightly lower in NIL-*GF14h^Arroz^* compared to Hitomebore, but a sufficient yield was guaranteed (Fig 5H and 5I). These results suggest that NIL-*GF14h^Arroz^* could be a valuable parental line for breeding via QTL pyramiding.

## Discussion

Here, we demonstrated that the functional *GF14h* allele present in Portuguese rice variety Arroz da Terra plays a pivotal role in supporting seed germinability under low-temperature

conditions. Although the regulation of seed germination by *GF14h* was recently documented, its activity under low-temperature conditions remained unclear [27,28]. While many genomic regions associated with LTG have been detected through QTL mapping and GWAS, only a few studies have identified the causal genes [3–20]. Indeed, LTG is a quantitative trait involving the cumulative effects of multiple genes and their epistatic relationships, making it difficult to assess the specific effect of a single genomic region on LTG. To eliminate the influence of other chromosomal regions on LTG, we first generated *qLTG11*-NIL containing only one of three QTLs detected in the Arroz da Terra background for genetic analysis. The analysis of *qLTG11*-NIL revealed that *GF14h* participates in LTG. Significantly, the NIL harboring the functional *GF14h* allele from Arroz da Terra in the Hitomebore background provides valuable pre-breeding materials for QTL pyramiding. These findings provide a genetic understanding of low-temperature germinability as well as new resources for rice breeding.

### Influence of *GF14h* on low-temperature germination

In this study, we identified *GF14h* as being implicated in LTG. In a previous study, a genetic complementation assay with a functional *GF14h* allele introduced into the rice cultivar Nipponbare background increased the germination rate at 30˚C, but only to a limited extent at 15˚C [28]. This result is not consistent with our finding that introducing functional *GF14h* into Hitomebore resulted in a significant improvement in germination at low temperature. Perhaps this discrepancy is due to differences in the rice varieties used in the germination assays. Notably, the haplotypes of *qLTG3-1*, a major QTL behind LTG [3], are different between Nipponbare and Hitomebore: whereas Nipponbare, which was released in 1961, harbors the functional allele of *qLTG3-1*, Hitomebore carries a loss-of-function allele with a deletion of 71 bp (S2 Fig) [37]. Moreover, the cultivars Koshihikari and Hayamasari, which carry the same loss-of-function *qLTG3-1* allele as the Hitomebore variety, were reported to exhibit lower germination rates at low temperatures than Nipponbare [37]. LTG tests using chromosome segment substitution lines derived from a cross between Koshihikari and Nipponbare indicated that *qLTG3-1* contributes to the difference in LTG between the two varieties [37]. Based on these observations, it is likely that Nipponbare has a better LTG ability than Hitomebore, which may have masked the effect of functional *GF14h* on LTG in the previous study [28]. Our study confirmed the involvement of *GF14h* in LTG through map-based cloning and analysis of knockout and overexpression lines. It is also worth mentioning that our experiments were performed in the *qLTG11*-NIL background, which allowed us to isolate the contribution of GF14h to LTG without any influence from *qLTG3-1* or other genes in the Arroz da Terra background. In summary, we provided multiple lines of evidence that *GF14h* contributes to LTG.

The expression pattern of functional *GF14h* during seed germination was previously unclear. While *GF14h* has been shown to be expressed in the aleurone layer surrounding the embryo [28], it is also highly expressed in the endosperm [27]. In the current study, we showed that *GF14h* was expressed throughout the seeds, but with a transient induction in expression in the endosperm at roughly the time of initiation of seed germination. GF14h was reported to regulate seed germination by interacting with the abscisic acid and gibberellin signaling pathways at optimal temperatures [27, 28]. We therefore suggest that GF14h controls LTG by interacting with various phytohormone signaling pathways.

### Low-temperature germinability in rice was lost due to selection in modern Japanese breeding

In this study, we performed haplotype network analysis of *GF14h* using many Japanese rice varieties. We identified ten distinct haplotypes based on 11 polymorphic sites in the *GF14h*

coding region. Of these, Hap4, encompassing the aus, indica, tropical japonica, temperate japonica, and *O. rufipogon* accessions, was defined at the center of the haplotype network. Furthermore, we determined that a 4-bp deletion converted the functional haplotype Hap2, which was derived from Hap4, into the nonfunctional haplotype Hap1. This finding is consistent with the relationship between Hap6 and Hap1 (which we defined as Hap2 and Hap1, respectively) observed by [27]. These results suggest that the nonfunctional allele represented by Hap1 was introduced into temperate *japonica* varieties from tropical *japonica* varieties carrying Hap2 and then spread to Japanese cultivars.

We studied the haplotype frequencies of *GF14h* and *qLTG3-1* in various Japanese landraces and cultivars, considering their time of release from breeding programs into the field. More than half of the Japanese landraces analyzed carried both functional *GF14h* and *qLTG3-1* alleles. However, the frequency of varieties carrying loss-of-function alleles for both *GF14h* and *qLTG3-1* has increased since crossbreeding began in the early 20th century. This trend has continued to the present, perhaps as a result of artificial selection to improve resistance to pre-harvest sprouting, with more than 80% of all varieties bred since 2001 carrying these loss-of-function alleles.

Our study provides a historical perspective on allelic shifts in Japanese rice breeding while highlighting the influence of modern breeding on genetic diversity. Further research is needed to elucidate the potential effects of higher frequencies of loss-of-function alleles on the overall phenotypic characteristics and ecological adaptability of Japanese rice varieties. In addition, as mentioned in previous reports [27,28], we believe that the reintroduction of functional alleles should be considered in order to develop cultivars suitable for labor-saving cultivation techniques such as direct seeding.

## Application to direct seeding for rice cultivation

Cultivation stability under direct seeding conditions is important for managing rice production costs and reducing labor. However, since rice is sensitive to low temperatures, improving seed germination and seedling establishment at low temperatures is a desirable breeding trait in high-latitude rice production areas such as Japan. In the current study, we developed a potentially useful NIL (NIL-*GF14h^Arroz^*) by introducing the functional *GF14h* allele into the Hitomebore background. Overexpressing this functional *GF14h* allele was previously shown to improve anaerobic germination and tolerance to seedling establishment under anaerobic conditions in laboratory experiments [27]. However, whether NIL-*GF14h^Arroz^* exhibits strong seedling vigor at low temperatures in rice fields remains to be determined. We previously identified a QTL associated with seedling vigor, *qPHS3-2* (QTL for plant height of seedling 3–2) [38]. *qPHS3-2* most likely corresponds to the gibberellin biosynthesis gene *GA20 oxidase1* (*OsGA20ox1*), a paralog of *Semi Dwarf1* (*sd-1*, corresponding to *OsGA20ox2*) [38]. Therefore, the pyramiding of *qPHS3-2* in NIL-*GF14h^Arroz^* by marker-assisted selection represents a promising approach for further improving seedling vigor in NIL-*GF14h^Arroz^*. On the other hand, enhancing LTG may conversely increase the risk of pre-harvest sprouting. Should the level of pre-harvest sprouting in NIL-*GF14h^Arroz^* pose practical issues, the pyramiding of QTLs for pre-harvest sprouting resistance, such as *Seed Dormancy 4* [39], may offer a solution.

In recent years, "early-winter direct seeding" has been experimentally tested as a new system of direct seeding for rice production in Japan [40]. In this system, seeds are directly sown in the early winter of the previous year instead of the spring. The sown seeds thus overwinter in snow-covered soil and germinate the following spring. The major advantage of this approach is that it can significantly decrease the amount of labor required by farmers during the busy spring season. However, it is challenging to overwinter the seeds of modern rice

varieties in the soil and achieve good seedling establishment [40–42]. We expect that reintroducing beneficial alleles like *GF14h^Arroz* that were lost during modern breeding into future rice varieties will enable the implementation of new cultivation practices and increase productivity.

## Materials and methods

### Plant materials

Rice was cultivated in a paddy field at Iwate Agricultural Research Center (39˚35'N, 141˚11'E). A recombinant inbred line (RIL) population of 200 $F_7$ lines was generated from a cross between Japanese variety Iwatekko and the high-LTG variety Arroz da Terra (S1 Fig) [31]. To develop NILs, rice cultivar Hitomebore was used as the recipient parent to generate NILs harboring the target genomic region from Arroz da Terra (S3A Fig), thereby establishing *qLTG3-2*-NIL, *qLTG11*-NIL, and NIL-*GF14h^Arroz* (Figs 1E, S3B and S4A).

### Evaluation of germination rate

Seeds for each line were harvested 45 days after heading, air-dried at 30˚C for two days, and stored at 4˚C until use. The seeds were air-dried at 50˚C for seven days in the dark to break dormancy. For germination tests, 50 seeds per replicate were incubated in a Petri dish filled with distilled water in the dark at 15˚C (low-temperature conditions) or 25˚C (optimal temperature conditions). For QTL-seq analysis, germination tests were conducted at 13˚C [31]. The germination rate was calculated as the total number of germinated seeds at each time point divided by the number of seeds tested. Seeds were considered to have germinated when the white coleoptile was visible.

To evaluate resistance to pre-harvest sprouting, panicles were harvested from NIL-*GF14h^Arroz* and Hitomebore 30 days after heading. The panicles were incubated in dark, wet conditions (by covering them with filter paper moistened with water) at 28˚C for 10 days, and seed germination was scored.

### QTL-seq analysis

LTG data for the 200 RILs derived from a cross between Iwatekko and Arroz da Terra (S1 Fig) were analyzed [31]. The top 20 RILs showing high-LTG and the bottom 20 RILs showing low-LTG phenotypes were selected to assemble the two bulk samples with contrasting LTG phenotypes. All seedlings with high or low LTG were pooled, and DNA was extracted from each bulk as previously described [31]. The genomic DNA of the two bulks was used to generate DNA-seq libraries and sequenced on a GAIIx sequencer (Illumina, CA, USA). QTL-seq was performed to identify QTLs related to LTG [31,32].

### Map-based cloning of *qLTG11*

To narrow down the *qLTG11* region, genotyping was performed using a cross population of *qLTG11*-NIL (BC$_2$F$_3$) backcrossed to Hitomebore. The germination rate under low-temperature conditions (15˚C) was measured to characterize LTG activity. High-resolution fine mapping with ten markers (markers B–K) between 23.466 Mb and 23.609 Mb on chromosome 11 identified six informative recombinants in the target region. Primers used for mapping are listed in S4 Table.

### *De novo* assembly of the Hitomebore and Arroz da Terra genomes

To reconstruct the *qLTG11* regions in Hitomebore and Arroz da Terra, *de novo* assembly was performed for each genome using Nanopore long reads and Illumina short reads according to

a published method [43]. To extract high-molecular-weight genomic DNA from leaf tissue for Nanopore sequencing, a NucleoBond high-molecular-weight DNA kit (MACHEREY-NAGEL, Germany) was used. Following DNA extraction, low-molecular-weight DNA was eliminated using a Short Read Eliminator Kit XL (Circulomics, MD, USA). Library preparation was then performed using a Ligation Sequencing Kit (SQK-LSK-109; Oxford Nanopore Technologies [ONT], United Kingdom) according to the manufacturer's instructions, and sequencing was performed using MinION (ONT, UK) for Arroz da Terra. For Hitomebore, Nanopore long reads sequenced by [43] were used. Base-calling of the Nanopore long reads was performed using Guppy 4.4.2 (ONT, UK). Sequences derived from the lambda phage genome were removed from the raw reads with NanoLyse v1.1.0 [44]. The first 50 bp of each read were then removed, as were reads with an average read quality score below 7 and reads shorter than 3,000 bases, using NanoFilt v2.7.1 [44]. The clean Nanopore long reads were assembled using NECAT v0.0.1 [45], setting the genome size to 380 Mb. To improve the accuracy of assembly, Racon v1.4.20 [46] was used twice for error correction using Nanopore reads, and Medaka v1.4.1 (https://github.com/nanoporetech/medaka) was subsequently used to correct misassembly. Two rounds of consensus correction were then performed using bwa-mem v0.7.17 [47] and HyPo v1.0.3 [48] with the Illumina short reads. Redundant contigs were removed using purge-haplotigs v1.1.1 [49], resulting in a *de novo* assembly of 374.8 Mb comprising 82 contigs for Hitomebore and 376.3 Mb consisting of 82 contigs for Arroz da Terra. The resulting genome sequences have been deposited in Zenodo (https://doi.org/10.5281/zenodo.10460309).

## Plant transformation

To generate *GF14h* knockout mutants, two single guide RNAs (sgRNAs) targeting exon 4 or exon 5 of *GF14h* were designed using the web-based service CRISPRdirect (crispr.dbcls.jp) [50] and cloned individually into the pZH::OsU6gRNA::MMCas9 vector [51]. The resulting vectors were introduced into Agrobacterium (*Agrobacterium tumefaciens*) strain EHA105 for transformation into *qLTG11*-NIL plants [52]. The target sites in the positive transformants were sequenced by Sanger sequencing to detect mutations. To obtain overexpression constructs, the full-length coding sequence of functional *GF14h* was amplified from total RNA extracted from *qLTG11*-NIL and cloned into the plant binary vector pCAMBIA1300 under the control of the cauliflower mosaic virus (CaMV) 35S promoter. The overexpression plasmid was introduced into Agrobacterium strain EHA105 for transformation of rice variety Hitomebore [52]. All primers used are listed in S4 Table.

## Expression analysis

Total RNA was extracted from germinating seeds using an RNA-suisui S kit (Rizo). Total RNA was treated with RNase-free DNase I (Nippon Gene). The resulting samples were reverse transcribed into first-strand cDNA using a PrimeScript RT Reagent Kit (Takara Bio). Quantitative PCR (qPCR) was conducted using a QuantStudio 3 system (Thermo Fisher Scientific) with Luna Universal qPCR Master Mix (New England Biolabs). The cycling parameters were 1 min at 95˚C, followed by 40 cycles of amplification (95˚C for 15 sec and 60˚C for 30 sec). The *Actin* gene (Os03g0718100) served as an internal control, and the Delta CT method was used to calculate the relative expression levels. The primer sets are listed in S4 Table.

## RNA-seq

Total RNA was extracted from Hitomebore and *qLTG11*-NIL seeds at 0, 1, 2, and 3 days after the onset of seed hydration under low (15˚C) or optimum (25˚C) temperature conditions

using an RNA-suisui S kit (Rizo, Ibaraki, Japan). Sequencing libraries were prepared using an NEBNext Ultra II Directional RNA Library Prep Kit for Illumina (New England Biolabs Japan, Tokyo, Japan) following the manufacturer's protocol. The libraries were sequenced in paired-end mode using an Illumina HiSeq X instrument (Illumina, CA, USA). The raw reads have been deposited in the DNA Databank of Japan (BioProject accession No. PRJDB17450; S5 Table). For quality control, reads shorter than 50 bases and those with an average read quality below 20 were discarded using Trimmomatic v0.36 [53], and poly(A) sequences were trimmed using PRINSEQ++ v1.2 [54]. The resulting clean reads were aligned to the *de novo* assembled Hitomebore and Arroz da Terra genomes with HISAT2 v2.1 [55]. BAM files were sorted and indexed with SAMtools v1.10 [56], and aligned reads were assembled into transcripts with StringTie [57] by combining bam files for each variety. In a similar manner, the expression data were generated using the Nipponbare reference genome downloaded from IRGSP-1.0 (https://rapdb.dna.affrc.go.jp/download/irgsp1.html).

### Haplotype network analysis

Sequencing datasets were obtained for 503 rice accessions. Of these, 379 were FASTQ files downloaded from the DNA Data Bank of Japan Sequence Read Archive (DRA) [33–35,58] and 124 were sequenced in this study (S2 Table). Details about DNA extraction, whole-genome sequencing techniques, and construction of the genotype datasets in VCF format are provided in a previous report [35]. This study specifically focused on the coding region of *GF14h*. Genotype information related to the coding region of *GF14h* was extracted from the VCF dataset. In addition, the *k*-mer analysis program (https://github.com/taitoh1970/kmer) [59] was used with Illumina short reads to detect the 4-bp deletion with high sensitivity. Genotype information for the presence of the 4-bp deletion was added to the VCF file, and 81 samples with heterozygous genotypes were discarded. A haplotype network was then constructed using the median-joining network algorithm [60] implemented in Popart v1.7 [61].

### Evaluation of agronomic traits and yield performance of NIL-*GF14h^Arroz^*

The grain yields of Hitomebore and NIL-*GF14h^Arroz^* were investigated in experimental paddy fields in 2023. Field experiments were conducted at the Iwate Agricultural Research Center (39˚35'N, 141˚11'E) in Kitakami, Iwate, Japan. A fertilization regime of $N:P_2O_5:K_2O = 6:6:6$ g $m^{-2}$ was applied as a basal dressing, and $N:K_2O = 2:2$ g $m^{-2}$ was applied as a top dressing. Seeds were sown in a seedling nursery box on 21 April, and seedlings were transplanted to the paddy field on 18 May. To evaluate agronomic traits, the seedlings were transplanted at a rate of one plant per hill, with a planting density of 22.2 hills $m^{-2}$. Culm length, panicle length, panicle number, grain number per plant, and grain weight per plant were measured at maturity. To evaluate yield performance, seedlings were transplanted with three plants per hill at a planting density of 16.7 hills $m^{-2}$. The $0.9 \times 5.0$ m experimental plots in the paddy fields were arranged in a randomized complete block design with three replicates. At maturity, 50 hills were harvested from each plot to measure brown rice yield. The hulls were removed using a rice huller (Model 25MC, Ohya Tanzo Factory Co., Ltd., Japan), and the hulled grains were screened with a grain sorter (1.9-mm sieve size). Brown rice yields were adjusted to 15% moisture content and converted to weight per hectare.

### Supporting information

**S1 Fig. Frequency distribution of germination rates in the RIL population and germination rates of selected RILs for QTL-seq analysis. (A)** Frequency distribution of germination rates at 13˚C after eight days from seed imbibition in 200 F7 RILs derived from a cross

between Iwatekko and Arroz da Terra [31]. **(B)** Selection of RILs with low cold germination rates. The 62 RILs with the lower germination rates from the first test (A) were tested for the second, and then the 20 RILs with the lowest germination rates were selected as a bulk sample for QTL-seq analysis. The bar graph shows the mean values of the two tests. **(C)** Selection of RILs with high cold germination rates. The 37 RILs with the higher germination rates from the first test (A) were tested for the second, and then the 20 RILs with the highest germination rates were selected for a bulk sample for QTL-seq analysis. The bar graph shows the average values of the two tests.
(PDF)

**S2 Fig. Multiple DNA sequence alignment of *qLTG3-1* variants.** Arroz da Terra and Italica Livorno harbor a functional *qLTG3-1* variant. Nipponbare carries another functional *qLTG3-1* variant due to the nonsynonymous substitution (*). Iwatekko, Hitomebore, and Hayamasari contain a loss-of-function variant for *qLTG3-1* due to a 71-bp deletion.
(PDF)

**S3 Fig. Generation of a near-isogenic line with high LTG in the Hitomebore background.**
**(A)** Strategy for the development of *qLTG3-2*-NIL, *qLTG11*-NIL, and NIL-*GF14h^Arroz^*. Molecular markers were used for foreground and background selection. **(B)** Diagram showing the genotype of NIL-*GF14h^Arroz^*. NIL-*GF14h^Arroz^* contains a 172-kb region on chromosome 11 harboring the Arroz da Terra allele of *GF14h*. Light blue bars indicate genomic fragments from Hitomebore; red bars indicate genomic fragments from Arroz da Terra.
(PDF)

**S4 Fig. Summary of *qLTG3-2*.** **(A)** Diagram showing the genotype of *qLTG3-2*-NIL containing the Arroz da Terra allele at *qLTG3-2* on chromosome 3. Light blue bars indicate genomic fragments from Hitomebore; red bars indicate genomic fragments from Arroz da Terra; dark blue bars indicate heterozygous regions. **(B)** Germination time courses for seeds of Hitomebore, the *qLTG3-2*-NIL, and Arroz da Terra at 15°C. Values are means ± SD of biologically independent samples (Hitomebore and NIL $n = 10$, Arroz da Terra $n = 5$).
(PDF)

**S5 Fig. Seed germination of *qLTG11*-NIL under optimal temperature conditions.** Germination time courses of seeds from Hitomebore and *qLTG11*-NIL at 25°C. Values are means ± SD of biologically independent samples ($n = 3$). Two-tailed t-test was used between *qLTG11*-NIL and Hitomebore for each time point (*$P < 0.05$ and **$P < 0.01$).
(PDF)

**S6 Fig. Comparison of the *qLTG11* genomic region in Hitomebore, Arroz da Terra, and Nipponbare.** Dot blot analyses of the genomic sequence in the *qLTG11* candidate region between **(A)** Hitomebore and Nipponbare and **(B)** Hitomebore and Arroz da Terra, using D-GENIES [62]. Based on the Nipponbare genome (IRGSP-1.0), the genomic region containing the causative gene is located at 23.512–23.564 Mb (approximately 52 kb) on chromosome 11. The genome sequence of Hitomebore is identical to that of Nipponbare. The candidate region corresponds to a fragment of approximately 94 kb in the Arroz da Terra genome.
(PDF)

**S7 Fig. Expression levels of the two annotated genes in the candidate genomic region of *qLTG11* based on RNA-seq data.** Total RNA was extracted from Hitomebore and *qLTG11*-NIL seeds at 0, 1, 2, and 3 days after the onset of seed imbibition under 15 or 25°C temperature conditions, followed by RNA-seq. The sequence reads were mapped to the Nipponbare genome (IRGSP-1.0), and expression data were obtained. **(A–B)** The expression levels of

Os11g0609600 (*GF14h*) during seed germination under 15˚C (A) and 25˚C (B) are shown. Data are presented as means ± SE. n = 3 biologically independent samples. **(C–D)** The expression levels of Os11g0609500 (*Jacalin-like lectin domain containing protein*) during seed germination under 15˚C (C) and 25˚C (D) are shown. Data are presented as means ± SE. n = 3 biologically independent samples.
(PDF)

**S8 Fig. Multiple DNA sequence alignment of *GF14h* variants.** Arroz da Terra carries a functional *GF14h* variant. Hitomebore and Nipponbare harbors a loss-of-function variant of *GF14h* due to a 4-bp deletion (black line).
(PDF)

**S9 Fig. CRISPR/Cas9-mediated genome editing of *GF14h*.** Top, diagram showing the *GF14h* locus, with the locations of the two sgRNA target sites marked by inverted red triangles. Bottom, sequencing results of putative *gf14h* mutants. The sgRNA target sites are underlined, and the PAMs are highlighted. The mutation sites in *GF14h*$^{Arroz}$ for the four mutants (*gf14h-1*, *gf14h-2*, *gf14h-3*, and *gf14h-4*) are indicated.
(PDF)

**S10 Fig. Effect of *GF14h* mutation on optimal-temperature germination. (A)** Representative photographs showing seed germination in wild-type harboring Arroz-type *GF14h* (WT$^{Arroz}$) and CRISPR/Cas9 knockout lines (*gf14-1*) at 3 days after the onset of seed imbibition. Scale bar, 1 cm. **(B)** Seed germination rate of WT$^{Arroz}$ and its CRISPR/Cas9 knockout lines at 2 days of seed imbibition at 25˚C. The two target constructs (S9 Fig) were introduced into the *qLTG11*-NIL line. Data are means ± standard error (WT$^{Arroz}$, *gf14h-1*, *gf14h-2* and *gf14h-4*, n = 3; *gf14h-3*, n = 2). Different lowercase letters indicate significant differences based on Tukey's HSD test ($P < 0.05$).
(PDF)

**S11 Fig. Relative *GF14h* expression levels in germinating seeds of *GF14h*$^{Arroz}$ overexpression lines and the parental line.** The *GF14h*$^{Arroz}$ overexpression construct was introduced into Hitomebore. *OsActin1* (Os03g0718100) was used for normalization. Values are means ± SE (*n* = 3 or 4). Different lowercase letters indicate significant differences based on Tukey's HSD test ($P < 0.001$).
(PDF)

**S12 Fig. Haplotype network of *GF14h*.** The *GF14h* genomic sequences obtained from 411 *O. sativa* varieties and 11 *O. rufipogon* accessions were used for analysis (S1 Table). The haplotype network was reconstructed by the median joining network algorithm [60] implemented in Popart v1.7 [61]. The haplotype Hap1 evolved from Hap2 by acquiring the 4-bp sequence, resulting in a nonfunctional *GF14h* gene. The Hitomebore cultivar contains Hap1 (nonfunctional), and the Arroz da Terra cultivar contains Hap9 (functional).
(PDF)

**S13 Fig. Pre-harvest sprouting of NIL-*GF14h*$^{Arroz}$.** Germination time courses of seeds from Hitomebore (blue circles) and the NIL-*GF14h*$^{Arroz}$ (pink triangles) under wet conditions at 28˚C. Seeds were harvested from tagged panicles 30 days after heading. Values are means ± SE of biologically independent samples (*n* = 3). The *P*-values calculated from *t*-tests at each time point are shown in the figure.
(PDF)

**S14 Fig. Development of a functional marker based on the 4-bp deletion in *GF14h*. (A)** Diagram of the sequence around the 4-bp InDel of *GF14h*. In the Hitomebore (loss-of-function) allele, the 4-bp deletion creates a SmlI restriction site. **(B)** Genotyping of the 4-bp deletion in *GF14h*. A genomic fragment containing the 4-bp InDel of *GF14h* was amplified by PCR and digested with SmlI. The products were separated on a 3% (w/v) agarose gel and stained with Midori Green. The PCR product from *GF14h^Arroz^* (approximately 500 bp) was not cleaved, whereas the PCR product from *GF14h^Hitomebore^* was cleaved, producing two fragments of approximately 250 bp each. Both bands were detected in heterozygous plants.
(PDF)

**S1 Table. Expression profile (TPM) during seed germination by RNA-seq.**
(CSV)

**S2 Table. List of rice varieties used in this study and their sequence read archive (SRA) IDs.**
(XLSX)

**S3 Table. Haplotypes of the *GF14h* gene analyzed in S12 Fig.**
(XLSX)

**S4 Table. Primers used in this study.**
(XLSX)

**S5 Table. List of RNA-seq samples used in this study and their sequence read archive (SRA) IDs.**
(XLSX)

**S6 Table. Numerical data used for the graphs.**
(XLSX)

## Acknowledgments

We thank the National Agriculture and Food Research Organization (NARO) gene bank, Japan, for providing rice seeds. This work was performed using the National Institute of Genetics (NIG) supercomputer at the Research Organization of Information and Systems (ROIS) National Institute of Genetics and the Academic Center for Computing and Media Studies (ACCMS) supercomputer at Kyoto University.

## Author Contributions

**Conceptualization:** Hiroki Takagi, Ryohei Terauchi, Akira Abe.

**Data curation:** Yusaku Sugimura, Akira Abe.

**Formal analysis:** Yusaku Sugimura, Yu Sugihara, Akira Abe.

**Funding acquisition:** Hiroki Takagi, Ryohei Terauchi, Akira Abe.

**Investigation:** Yusaku Sugimura, Kaori Oikawa, Yu Sugihara, Hiroe Utsushi, Eiko Kanzaki, Kazue Ito, Yumiko Ogasawara, Tomoaki Fujioka, Hiroki Takagi, Motoki Shimizu, Hiroyuki Shimono, Akira Abe.

**Supervision:** Ryohei Terauchi, Akira Abe.

**Validation:** Yusaku Sugimura, Akira Abe.

**Visualization:** Yusaku Sugimura, Yu Sugihara, Akira Abe.

**Writing – original draft:** Yusaku Sugimura, Akira Abe.

**Writing – review & editing:** Yusaku Sugimura, Ryohei Terauchi, Akira Abe.

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
