## [Decision Letter · Decision Letter 0]

16 Apr 2024

Dear Dr Abe,

Thank you very much for submitting your Research Article entitled 'Impact of rice *GENERAL REGULATORY FACTOR14h* (*GF14h*) on low-temperature seed germination and its application to breeding' to PLOS Genetics.

First, apologies for the unusually long processing period. We have tried hard to obtain a third independent review, unfortunately without success. However, as both submitted reviews appreciate the importance of the topic but recommend a substantial revision of the manuscript, we decided to make their comments available to you now, so that you can consider them in a revised version. Based on the reviews, we will not be able to accept this version of the manuscript, but we would be willing to review a much-revised version. We cannot, of course, promise publication at that time.

As you will see from the reviewer’s comments, they found discrepancies between experimental details, incomplete data sets, and vague statements when precise quantification is needed. Differences in germination behavior at certain conditions should consider differences other than only the modified parameter, and there are more complex ways to characterize germination kinetics. In addition, if referring to transcriptome data, these need to be made available, and expression data should be quantitative. Please discuss with your co-authors if you can give more information on the other potentially relevant genes, or on other regulatory components than ABA and GA. We hope that these comments will help to prepare a substantially revised version.

If you decide to revise the manuscript for further consideration at PLOS Genetics, please aim to resubmit within the next 60 days, unless it will take extra time to address the concerns of the reviewers, in which case we would appreciate an expected resubmission date by email to plosgenetics@plos.org.

We are sorry that we cannot be more positive about your manuscript at this stage. Please do not hesitate to contact us if you have any concerns or questions.

Yours sincerely,

Ortrun Mittelsten Scheid

Academic Editor

PLOS Genetics

Claudia Köhler

Section Editor

PLOS Genetics

Reviewer's Responses to Questions

**Comments to the Authors:**

Reviewer #1: Direct seeding of rice is becoming popular in recent years. However, this approach requires varieties with vigorous low temperature germination especially when sown under cold conditions in the early spring. This manuscript revealed the candidate gene GF14h regulates low temperature germination in rice. The authors also investigated the distribution of functional GF14h allele in the popular varieties during modern breeding. Further, the contribution of functional GF14h allele from Arroz da Terra was confirmed for rice direct seeding in cold regions. Several comments are listed below.

1. It has been revealed that GF14h regulates rice seed germination through abscisic acid and gibberellin signaling pathways. In this study, the mechanism of GF14h influencing low temperature germination was still unclear. Whether other novel factors, except abscisic acid and gibberellin signaling, involved in the GF14h medicated regulation of low temperature germination?

2. The germination rates of RIL population were determined at 8 days of seed imbibition at 13°C. But 15°C was used for NIL, knockout and overexpression lines. Why?

3. Results showed that qLTG11-NIL seeds also germinated more rapidly than Hitomebore seeds under normal conditions (25ºC). For other knockout and overexpression lines, how about the germination phenotype under normal conditions (25ºC)? The difference of germination speed will influence the low temperature germination?

4. NIL-GF14hArroz was slightly more susceptible to pre-harvest sprouting than Hitomebore. It is also a challenge for the future rice breeding.

5. Whether GF14h regulates the seedling establishment or seedling growth at low temperatures?

Reviewer #2: The review is uploaded as attachment.

**Have all data underlying the figures and results presented in the manuscript been provided?**

Reviewer #1: None

Reviewer #2: **No: **As mentioned in the written review, the manuscript strongly misses the RNAseq data.

PLOS authors have the option to publish the peer review history of their article (what does this mean?). If published, this will include your full peer review and any attached files.

Reviewer #1: No

Reviewer #2: No

---

## [Decision Letter · Decision Letter 1]

12 Jul 2024

Dear Dr Abe,

We are pleased to inform you that your manuscript entitled "Impact of rice *GENERAL REGULATORY FACTOR14h* (*GF14h*) on low-temperature seed germination and its application to breeding" has been editorially accepted for publication in PLOS Genetics. Congratulations!

Please make sure to check the file format in the Supplement, as requested by reviewer 2. You may also follow the suggestion of reviewer 1 for a more detailed discussion of the endosperm-specific difference in gene expression. In both cases, this will not need to go back to the reviewers.

Yours sincerely,

Ortrun Mittelsten Scheid

Academic Editor

PLOS Genetics

Claudia Köhler

Section Editor

PLOS Genetics

Comments from the reviewers (if applicable):

Reviewer's Responses to Questions

**Comments to the Authors:**

Reviewer #1: Thanks for the authors' careful revisions. All my concerns have been well addressed. I have no other major comments. In Figure 2, the interesting result is that the GF14h is mainly expressed in endosperm while not in embryo. If possible, the authors can add more discussions on the potential mechanism of GF14h on seed germination.

Reviewer #2: Thank you to the Authors for answering all the major and minor issues, all my questions were answered and explained, I don’t have further questions.

I believe that the addition of germination characteristics of the varieties and the RIL lines with statistical analysis and the RNAseq data is useful and important, making the manuscript suitable for publication.

I have only one minor technical comment regarding Supplemental Table 1; after decompressing the file, on my computer the table does not separate columns. I ask the Authors to double-check the file format for a user-friendly presentation.

**Have all data underlying the figures and results presented in the manuscript been provided?**

Reviewer #1: None

Reviewer #2: Yes

PLOS authors have the option to publish the peer review history of their article (what does this mean?). If published, this will include your full peer review and any attached files.

Reviewer #1: No

Reviewer #2: No

**Data Deposition**

http://datadryad.org/submit?journalID=pgenetics&manu=PGENETICS-D-24-00232R1

**Press Queries**

---

## [Editor Report · Acceptance letter]

1 Aug 2024

PGENETICS-D-24-00232R1 

Impact of rice *GENERAL REGULATORY FACTOR14h* (*GF14h*) on low-temperature seed germination and its application to breeding 

Dear Dr Abe, 

We are pleased to inform you that your manuscript entitled "Impact of rice *GENERAL REGULATORY FACTOR14h* (*GF14h*) on low-temperature seed germination and its application to breeding" has been formally accepted for publication in PLOS Genetics! Your manuscript is now with our production department and you will be notified of the publication date in due course.

With kind regards,

Dorothy Lannert

PLOS Genetics

On behalf of:
